# Defining Potentially Inappropriate Prescriptions for Hypoglycaemic Agents to Improve Computerised Decision Support: A Study Protocol

**DOI:** 10.3390/healthcare9111539

**Published:** 2021-11-11

**Authors:** Paul Quindroit, Nicolas Baclet, Erwin Gerard, Laurine Robert, Madleen Lemaitre, Sophie Gautier, Chloé Delannoy-Rousselière, Bertrand Décaudin, Anne Vambergue, Jean-Baptiste Beuscart

**Affiliations:** 1Univ. Lille, CHU Lille, ULR 2694-METRICS: Évaluation des Technologies de Santé et des Pratiques Médicales, F-59000 Lille, France; baclet.nicolas@ghicl.net (N.B.); erwin.gerard.etu@univ-lille.fr (E.G.); laurine.robert@univ-lille.fr (L.R.); jean-baptiste.beuscart@univ-lille.fr (J.-B.B.); 2Lille Catholic Hospitals, Department of Infectious Diseases, F-59000 Lille, France; 3CHU Lille, Institut de Pharmacie, F-59000 Lille, France; chloe.rousseliere@chru-lille.fr (C.D.-R.); bertrand.decaudin@chru-lille.fr (B.D.); 4CHU Lille, Department of Diabetology, Endocrinology, Metabolism and Nutrition Lille University Hospital, F-59000 Lille, France; madleen.lemaitre.etu@univ-lille.fr (M.L.); anne.vambergue@chru-lille.fr (A.V.); 5Lille University, School of Medicine, F-59000 Lille, France; 6CHU Lille, Centre Régional de Pharmacovigilance, F-59000 Lille, France; sophie.gautier@chru-lille.fr; 7Univ. Lille, CHU Lille, ULR 7365-GRITA: Groupe de Recherche sur les Formes Injectables et les Technologies Associées, F-59000 Lille, France; 8European Genomic Institute for Diabetes, University School of Medicine Lille, F-59000 Lille, France

**Keywords:** study protocol, type 2 diabetes, hypoglycaemic agents, potentially inappropriate prescriptions, inappropriate prescribing, methodology research, decision support systems

## Abstract

In France, around 5% of the general population are taking drug treatments for diabetes mellitus (mainly type 2 diabetes mellitus, T2DM). Although the management of T2DM has become more complex, most of these patients are managed by their general practitioner and not a diabetologist for their antidiabetics treatments; this increases the risk of potentially inappropriate prescriptions (PIPs) of hypoglycaemic agents (HAs). Inappropriate prescribing can be assessed by approaches that are implicit (expert judgement based) or explicit (criterion based). In a mixed, multistep process, we first systematically reviewed the published definitions of PIPs for HAs in patients with T2DM. The results will be used to create the first list of explicit definitions. Next, we will complete the definitions identified in the systematic review by conducting a qualitative study with two focus groups of experts in the prescription of HAs. Lastly, a Delphi survey will then be used to build consensus among participants; the results will be validated in consensus meetings. We developed a method for determining explicit definitions of PIPs for HAs in patients with T2DM. The resulting explicit definitions could be easily integrated into computerised decision support tools for the automated detection of PIPs.

## 1. Introduction

In France, in 2017, 5% of the general population (more than 3.3 million people) were taking drug treatments for diabetes mellitus (DM) [1]. Around 90% of these individuals have type 2 diabetes mellitus (T2DM), which is characterised by insulin resistance [2]. Patients with diabetes typically have several comorbidities and a high prevalence of polypharmacy, which increases the risk of drug–drug interactions and poor treatment adherence [3]. Detournay et al. estimated that the annual cost of hospitalisations for hypoglycaemia in type 2 diabetic patients in France was between 80 and 105 million euros [4]. Over the last decade, an increase in the number of available treatment options has complicated the management of patients with diabetes. In France, 87% of the patients with T2DM are managed by a general practitioner alone and do not consult a diabetologist for their DM treatments [3]. The increasingly complex management of T2DM can be challenging for most physicians and might, therefore, increase the risk of potentially inappropriate prescriptions (PIPs) of hypoglycaemic agents (HAs) among patients with T2DM [5,6,7].

Inappropriate prescribing is defined as the potential use of medicine with more risks than benefits—particularly when safer alternatives are available. It includes (i) the prescription or use of more drugs than are clinically needed (i.e., overuse); (ii) the incorrect prescription or use of drugs that are needed (i.e., misuse); (iii) the failure to prescribe or use drugs that are needed (i.e., underuse) [8,9,10,11]. Inappropriate prescribing is associated with increased morbidity and mortality rates and has major financial consequences by triggering hospital admission or prolonging hospital stays [12].

Inappropriate prescribing can be identified through two very distinct approaches. The implicit approach is based on an expert judgement of the quality of care with regard to the patient’s condition and the medical literature [13,14]. On the contrary, the explicit approach is based on prescription data and does not require expert assessment (i.e., can be directly implemented for use in medical informatics) [13,14].

In this regard, inappropriate prescribing of HAs to patients with T2DM has mainly been reported based on an implicit approach [15,16,17,18]. Other studies developed prescribing quality indicators (PQIs) for assessing the quality of prescribing in patients at the population level [19,20,21]. These PQIs can be used to study populations but cannot be used in everyday medical clinical decision support systems at the patient level. Consequently, AL-Musawe et al. recently highlighted the lack of studies addressing the serious clinically relevant drug–drug interactions and PIPs in the elderly with T2DM [22].

We reasoned that the use of explicit definitions of PIPs of HAs might be of value in this context. Although an explicit approach is rarely applied to T2DM, it is frequently used in other areas of medicine (e.g., geriatrics). For example, several guidelines (e.g., the Beers and the STOPP/START criteria) provide explicit definitions of PIPs in older people [23,24]. Nevertheless, these guidelines provide only five criteria of HAs that concern only three oral antidiabetics (i.e., biguanides, sulphonylureas, and thiazolidinediones) and cannot cover all PIPs of HAs for T2DM.

It is known that explicit definitions (i) increase the prescribers’ awareness of PIPs, (ii) enable the automated detection of PIPs of HAs in electronic health records, (iii) underpin the development of clinical decision support systems, and (iv) generate important public health data [8,25,26,27]. To the best of our knowledge, explicit definitions of PIPs of HAs in patients with T2DM have not previously been listed.

## 2. Materials and Methods

### 2.1. Aim, Scope, and Steering Committee

The objective of the present study protocol is to describe a method to develop explicit definitions of PIPs of HAs in patients with T2DM. We shall deliberately limit the scope of our study to T2DM and will exclude type 1 diabetes mellitus, gestational diabetes, other types of diabetes, patient education, and advice for patients on the complications of diabetes. Our study focuses only on HAs prescribed to T2DM patients and excludes other treatments that are prescribed to prevent complications among T2DM patients (e.g., antihypertensive drugs or acetylsalicylic acid).

A steering committee (comprising a diabetologist, a general practitioner, a clinical pharmacist, a community pharmacist, and a pharmacologist) is being set up to validate the study’s methodology and monitor its progress.

### 2.2. Ethics Approval

The qualitative study and the Delphi survey (steps 2 and 3) will include only health professionals in order to obtain their expert opinion. Within the framework of French legislation, this type of study does not concern the ‘Jardé law’ and does not require the opinion of an ethics committee [28]. The data collected will be declared to the National Commission on Informatics and Liberty (CNIL) in accordance with French and European regulations (General Data Protection Regulation).

### 2.3. Preliminary Search

Before starting to develop explicit definitions of PIPs for HAs in patients with T2DM, we checked that there were no publications on this subject by searching the Medline via PubMed, the Cochrane Library, and the International Prospective Register of Systematic Reviews (PROSPERO) databases up until August 2021. We did not find any publications on this subject.

### 2.4. Overview of the Work to Be Performed

The explicit definitions will be developed in three steps (Figure 1). This mixed method will contribute to addressing the objective of the study. The systematic review and the qualitative study will aim to identify as many explicit definitions as possible, while the Delphi survey will aim to provide a consensus among them.

### 2.5. Step 1: The Systematic Review

#### 2.5.1. Purpose

The objective of the systematic review is to identify all published explicit definitions of PIPs for HAs in patients with T2DM. The results will be reported in compliance with the Preferred Reporting Items for Systematic Reviews and Meta-Analyses (PRISMA) statement recommendation. We will systematically search several databases, including Medline via PubMed, Web of Science, Scopus, and Embase. The systematic review’s flow diagram is shown in Figure 2; the study is registered with the PROSPERO (CRD42021250028).

#### 2.5.2. Study Selection

We will consider studies that investigate explicit definitions of PIPs for HAs in patients with T2DM (ATC code A10B). The following will be excluded: studies published before 2010, studies that included patients with type 1 diabetes mellitus, gestational diabetes, or other types of diabetes, studies of diabetes-related complications (macroangiopathy and microangiopathy), studies of patients with T2DM but that do not consider HAs, publications with implicit definitions (i.e., clinical judgment), animal studies, publications that are not written in French or English, and publications for which the abstract and the full text are not available.

#### 2.5.3. Search Strategy

The search strategy combines three classes of search terms: ‘type 2 diabetes mellitus’ AND ‘hypoglycaemic agents’ AND ‘potentially inappropriate prescriptions’. For example, Baclet et al. found 30 search terms for ‘potentially inappropriate prescriptions’ [27]. Our search strategy will be developed with help of a scientific librarian. The search strategy for Medline via PubMed is presented in Appendix A (Table A1). Only published studies identified by this search will be included in the systematic review. Two investigators (P.Q. and E.G.) will independently review the title, abstracts, and full texts of the publications identified in the search. A third investigator (M.L.) will be called upon to resolve any differences of opinion, if needed.

#### 2.5.4. Data Extraction

The search results will be exported to the Zotero bibliography manager, in order to store, manage, and organise the obtained bibliographical references. The following characteristics (and others) will be extracted from the selected publications: the type of study, the year of publication, the country, the population, the type of HA, and the explicit definitions. All data will be extracted by two independent reviewers (P.Q. and E.G.). Again, a third investigator (M.L.) will be called upon to resolve any differences of opinion, if necessary. Full manuscripts will be obtained for all titles and abstracts that met the inclusion criteria and will be coded in NVivo 12 version software (QSR International Pty Ltd. Australia 2020).

#### 2.5.5. Data Analysis

The data extracted by the two reviewers (P.Q. and E.G.) will be merged to create a single list of explicit definitions. In fact, some articles might contain different written formulations of the same explicit definition of a PIP. Hence, the explicit definitions extracted by the two reviewers will be renamed so that they are as similar as possible to the extracted data. Similar definitions from different publications will be grouped together. Again, a third investigator (M.L.) will be called upon to resolve any differences of opinion, if necessary. The definitions will be classified according to organs (e.g., kidneys, liver, pancreas, heart, etc.) and HAs according to the ATC classification system.

### 2.6. Step 2: The Qualitative Study

#### 2.6.1. Purpose

A qualitative study may help to collate expert definitions before the Delphi survey. Similar approaches are often used in other areas of medicine, such as the development of a core outcome set [29,30]. Here, we intend to conduct a qualitative study and thereby complete the definitions identified in the systematic review. The study will comply with the Consolidated Criteria for Reporting Qualitative Studies (COREQ).

#### 2.6.2. Participants

We will organise two focus groups, each of which will feature between 6 and 10 experts in the prescription of HAs: diabetologists, general practitioners, clinical pharmacists, community pharmacists, and pharmacologists. There will be no contact between the researchers and the focus group members before the group meets. The participants’ characteristics will be recorded: age, sex, year of qualification, medical specialties, and the type of practice (general hospitals, university hospitals, or private practice). Participants will be recruited via an e-mail invitation.

#### 2.6.3. The Focus Groups

Two investigators will be present: a facilitator (E.G.) and an observer (P.Q.); the latter will note relevant additional items. No other people will be present. All participants will be required to consent to the meeting being videoed and audiotaped. First, each participant will be asked to list the HAs that they feel are worthy of consideration. Next, PIPs for each HA will be discussed. The facilitator (E.G.) will not influence or structure the participants’ discussions in any way but may help to refocus the discussion on the group’s objective. After each focus group meeting, the steering committee will meet to adjust the procedure and the meeting guide, if needed.

#### 2.6.4. Data Extraction

The audio recording of each group’s discussion will be transcribed word for word. Textual discourse analysis will be conducted independently by two investigators (P.Q. and E.G.). The objective is to identify all verbatim elements that refer to explicit definitions of PIPs for HAs in patients with T2DM. Any disparity between the two investigators’ respective analyses will be discussed, resolved by consensus, and then validated by the steering committee.

#### 2.6.5. Data Analysis

Each verbatim referring to an explicit definition will be independently analysed by two investigators (P.Q. and E.G.) using NVivo software (version 12, QSR International, Melbourne, Australia). The objective is to group together the textual elements that refer to the same definition. For each definition, each investigator will suggest a formulation that is as close as possible to the verbatim. The same definitions from each verbatim will be grouped together. The definitions will finally be classified according to organs (e.g., kidneys, liver, pancreas, heart, etc.) and HAs according to the ATC classification system. Any differences in formulation or grouping will be discussed by the two investigators (P.Q. and E.G.), resolved by consensus, and then systematically validated by the steering committee and a third party (e.g., diabetologists or general practitioners who are not participating in a focus group). The list of explicit definitions will then be submitted to all the focus group participants for final validation.

### 2.7. Step 3: Preparation of the Delphi Survey

#### 2.7.1. Purpose

The purpose of the Delphi survey is to gather opinions, build consensus among experts, and reduce the number of explicit definitions to a priority list. The initial preparation and the selection of experts are key factors in performing a Delphi survey.

#### 2.7.2. Preparation and Validation of the List and Recruitment of Key Participants

The results of the systematic review and qualitative research are merged into a single list of explicit definitions by expert clinicians and researchers. The development of a list of definitions for the Delphi survey is summarised in Figure 3.

Two investigators (P.Q. and E.G.) will independently prepare the list of definitions, in a two-step process. In the first step, definitions from the systematic review of the English literature will be translated into French. The two investigators (P.Q. and E.G.) will check that the French translation is representative of the English verbatim associated with each definition. It should be borne in mind that formulations of the definitions will have been validated by the two investigators (P.Q. and E.G.) during the systematic review. In the second step, each of the two investigators (P.Q. and E.G.) will compare the list from the systematic review (translated into French) with the list from the qualitative study (in French). The goals are to (i) identify and group together similar definitions from the two lists and (ii) identify and exclude explicit definitions that do not fit the scope of the study. Any disagreement will be resolved by discussion and consensus between the two researchers (P.Q. and E.G.) and then validated by the steering committee.

Lastly, a group of experts will check that each definition in the final list corresponds to the definitions from the systematic review and from the qualitative study. Any issues identified by the group of experts will be resolved through discussion and consensus at a joint meeting of the participants, with facilitation by the two researchers (P.Q. and E.G.). The list of explicit definitions will then be submitted to the steering committee for final validation.

The panel size is known to have a significant influence on the relevance and reliability of Delphi survey results. To limit variability, a panel of 100 experts appears to be optimal [31]. Experts in the area of HA prescription will be recruited: diabetologists, general practitioners, clinical pharmacists, community pharmacists, and pharmacologists. The characteristics and the number of each category of participants will be calculated to provide a balance between the specialists represented on the panel. The expected composition is shown in Table 1.

### 2.8. Methods for the Delphi Survey

#### 2.8.1. The Goupile Tool

The Delphi survey will be developed using the Goupile tool: an open-source electronic data capture application for easy form creation and data entry. The tool involves (i) an online questionnaire for participants that includes medical terms, plain language terms, and their explanations; and (ii) an option for suggesting new definitions or commenting on existing definitions. Each participant is blinded to the other participants’ identities and answers.

#### 2.8.2. Consensus

In each cycle of the Delphi survey, the participants will rate each of the definitions on a 1–9 scale, according to the Grading of Recommendations Assessment, Development and Evaluation (GRADE) method [32,33]. A rating of 7–9 indicates definitions of critical importance, a rating of 4–6 indicates definitions that are important but not critical, and a rating of 1–3 indicates definitions of limited importance.

We will consider a consensus on including a given definition to have been reached if 70% or more of the participants consider the definition to be critically important (a rating of 7–9) and 15% or less of the participants consider it to be of limited importance (a rating of 1–3).

### 2.9. Running the Delphi Survey

In line with the French legislation on evaluations of professional practice, the Delphi survey does not require approval by an independent ethics committee [25]. Nevertheless, participants will be required to give their written consent to participation, and data collected will be declared to the National Commission on Informatics and Liberty (CNIL) in accordance with French and European regulations (General Data Protection Regulation).

#### 2.9.1. Maximising the Response Rate

Participants will receive an e-mail message with a link to the survey via the Goupile tool. The survey will remain online for 3 weeks, and reminder emails will be sent every 7 days after the initial invitation. One member of the steering committee for each specialty will be responsible for overseeing the follow-up and deadlines.

#### 2.9.2. Rounds 1 to 3

Round 1: Participants will rate each definition, according to the GRADE method. Participants may also suggest additional definitions and comment on their rankings. The definitions suggested by participants will be checked by two investigators and discussed with the steering committee. Definitions will be considered very important if 70% or more participants rate them 7–9. Definitions will be considered of limited importance if 70% or more participants rate them 1–3.

Consensus Meeting 1: All participants will be invited to participate in a virtual consensus meeting. Numerical data on the results of Round 1 will be provided. The final validation of the definitions that receive a rating of 7–9 from 70% or more of the participants will be discussed and voted on, if necessary. The definitive exclusion of definitions that have received a rating of 1–3 by 70% or more of the participants will be discussed and voted on, if necessary. All definitions not excluded or not validated will be submitted to Round 2.

Round 2: Only participants who participated in Round 1 will be invited to participate in Round 2. Definitions not excluded or not validated in consensus meeting 1 will be submitted, along with any new definitions suggested by the participants in Round 1. Participants will see the ratings and the median score given to each definition during Round 1.

Consensus Meeting 2: All participants in Round 2 will be invited to participate in a virtual consensus meeting. Numerical data on the results of Round 2 will be provided. The final validation of the definitions that receive a rating of 7–9 from 70% or more of the participants will be discussed and voted on, if necessary. The definitive exclusion of definitions that have received a rating of 1–3 by 70% or more of the participants will be discussed and voted on, if necessary. Definitions not excluded or not validated will be submitted to Round 3.

Round 3 (if necessary): Only participants who participated in Round 2 will be invited to participate in Round 3. All participants will be asked to answer YES or NO for the inclusion of each definition in the final list. The participants will be specifically encouraged not to rate YES for all presented definitions.

#### 2.9.3. The Final Consensus Meeting

The final step is a consensus meeting between the participants having completed all three Rounds and the members of the steering committee. The meeting will discuss, review, and vote on a final list of explicit definitions. The results of the three Rounds will be presented and discussed. The members of the steering committee will be sent the results of Round 3 of the Delphi prior to this meeting, to give them time to consider their answers and those of the other participants. The meeting will be chaired by a non-voting investigator. After a discussion of each explicit definition, the participants will vote in a secret ballot. Prior to the establishment of the final consensus, the list of explicit definitions will be reviewed and finalised.

## 3. Discussion

The objective of the study protocol is to develop explicit definitions of PIPs for HAs in patients with T2DM. The development of the explicit definitions will be achieved in a three-step process: a systematic review, a qualitative study, and a Delphi survey. We expect the resulting explicit definitions to help optimise the management of patients with T2DM. The definitions have the potential to reduce unwarranted variations in practice, improve the quality and safety of healthcare, and generate important public health data [8,25,26,27]. Nevertheless, the method is exploratory and will probably not generate an exhaustive list of explicit definitions and some of the definitions might be limited in scope. Explicit criteria must also be regularly updated in line with evolving clinical evidence. Since diabetes is a complex disease with heterogeneous populations, and as the majority of diabetics are type 2 and are followed by general practitioners alone, our work focuses on HAs in patients with T2DM.

We expect that the study’s results will contribute significantly to the use of an explicit approach by improving the quality of explicit definitions of PIPs for HAs in patients with T2DM. This type of definition could be easily integrated into computerised decision support tools for the automated detection of PIPs and the re-evaluation by a clinical pharmacist [34]. It has been shown that rule-based clinical decision support systems that provide patient-specific recommended care protocols are associated with better clinical outcomes for patients.

## Figures and Tables

**Figure 1 healthcare-09-01539-f001:**
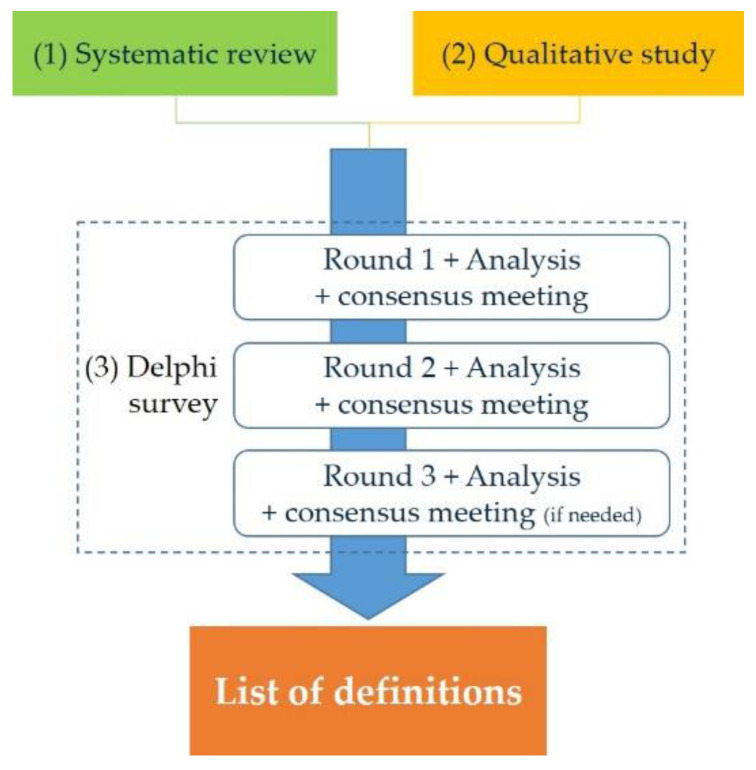
The three-step approach used to develop explicit definitions.

**Figure 2 healthcare-09-01539-f002:**
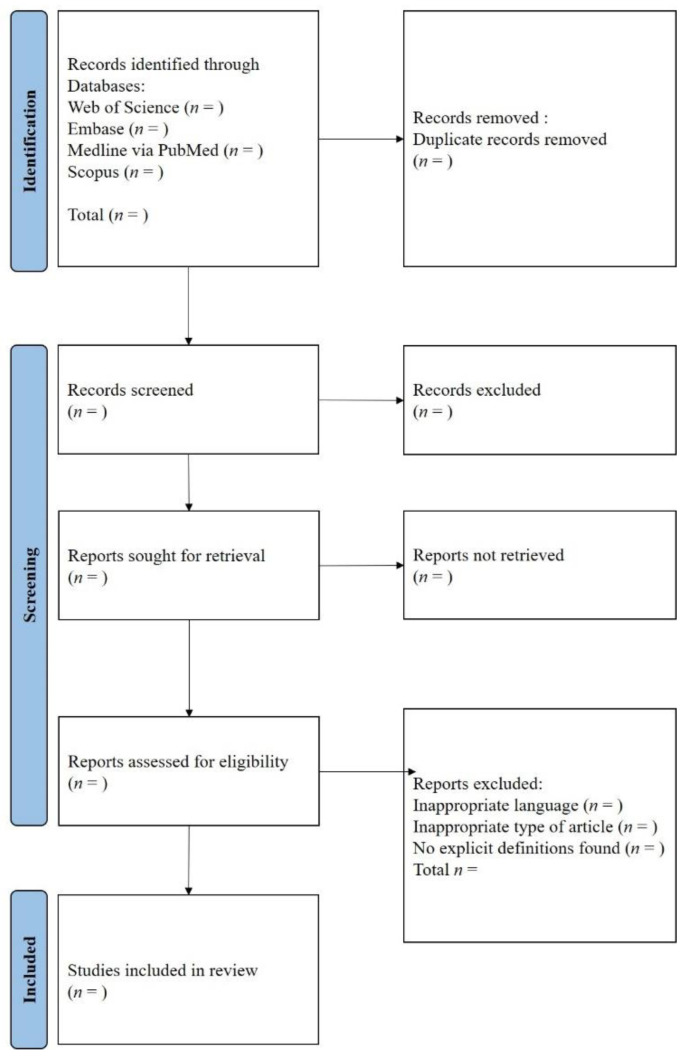
PRISMA flow diagram of the literature screening and selection process.

**Figure 3 healthcare-09-01539-f003:**
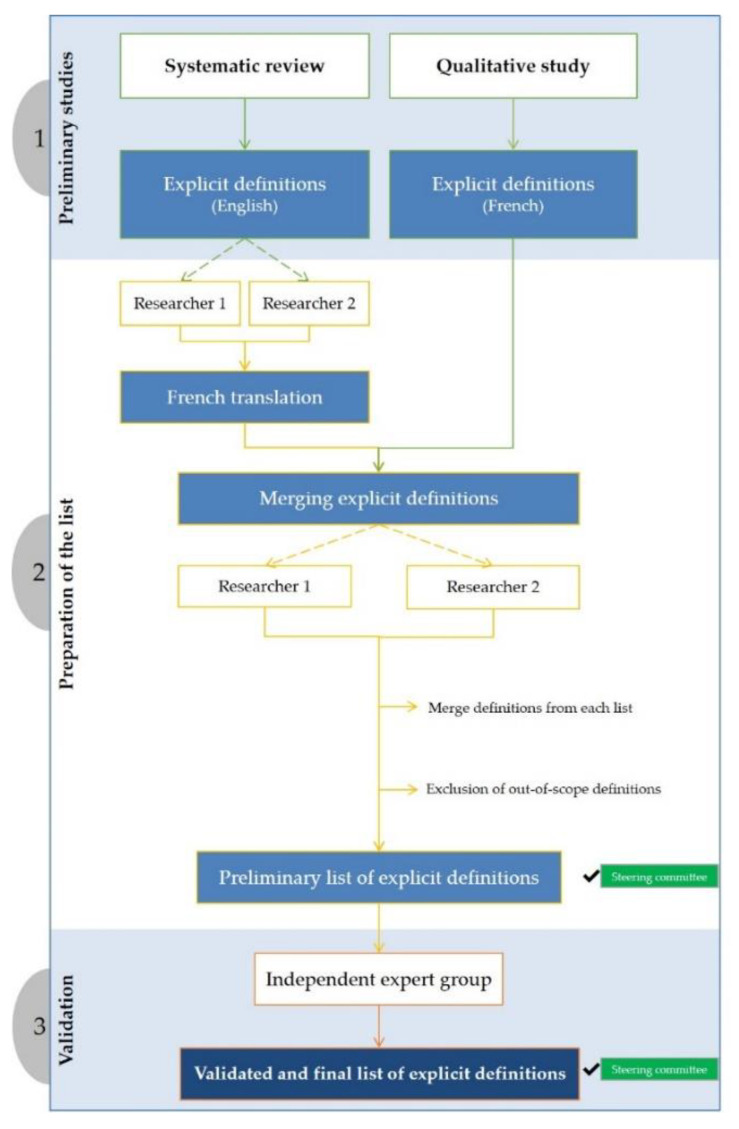
Method for preparing a list of explicit definitions of PIPs for the Delphi survey.

**Table 1 healthcare-09-01539-t001:** Expected distribution of participants.

Medical Speciality	Participants (*n* Total = 100)
Diabetologists	*n* total = 40
University hospital practitioners	*n* = 15
General hospital practitioners	*n* = 15
Private practitioners	*n* = 10
Outpatient care	*n* total = 30
General practitioners	*n* = 15
Community pharmacists	*n* = 15
Other specialists	*n* total = 30
Clinical pharmacists	*n* = 15
Pharmacologists	*n* = 15

## Data Availability

Not applicable.

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
