# Peer review of "Defining Potentially Inappropriate Prescriptions for Hypoglycaemic Agents to Improve Computerised Decision Support: A Study Protocol"

_healthcare, 2021, doi:10.3390/healthcare9111539_

Round 1
Reviewer 1 Report
Accept the study protocol with minor revision.

Reviewer 2 Report
The authors propose a study protocol for the development of explicit definitions of potentially inappropriate prescriptions for glucose-lowering drugs in patients with type 2 diabetes.
The paper is well written, and the protocol study is well designed. I only have a suggestion for the authors, table 1 information should be in an annex.
Reviewer 3 Report
This is a methods paper to determine the criteria for inappropriate prescribing of glucose-lowering agents used in patients with diabetes mellitus type 2. The following suggestions and comments are offered in order to strengthen an already strong manuscript.
Title: Quite long. Consider changing it to: Defining Potentially Inappropriate Prescriptions for Hypoglycemia Agents to Improve Computer Decision Support: A Study Methodology
Abstract: Change "glucose-lowering" to "hypoglycemic" and GLAs to HAs throughout the manuscript to be more scientific. Consider adding "mixed methods" to the description of your study, as "In a mixed methods, multistep process," at line 26.
Key words: "Potentially inappropriate prescriptions," "glucose-lowering agents," and "study protocol" are not MeSH terms. Consider changing them to "prescribing, inappropriate," "hypoglycemic agents," and "methodology, research" instead. Add "decision support systems, clinical" to your list. These changes would improve the searchability of your manuscript. (A quick PubMed search comparing the search results from RCTs glucose-lowering agents (638 citations) and hypoglycemic agents (19,126 citations) over the last twenty years showed that hypoglycemic agents (HA) is the preferred term. You use HA in your search strategy #2, so I would use that term throughout your manuscript.)
Introduction: At line 49, a reference is needed that would serve as the basis for the statement, "The increasingly complex management of T2DM can be challenging for most physicians and might therefore increase the risk of potentially inappropriate prescriptions (PIPs) of glucose-lowering agents (GLAs) among patients with T2DM."
Line 67: typo PQL instead of QPL (English abbreviation)
Methods: The study protocol uses PRISMA guidelines for reporting systematic reviews and meta-analyses, and is registered at PROSPERO. Grading of Recommendations Assessment, Development and Evaluation (GRADE) method will be used to achieve consensus for identifying critical definitions.
At line 147, why wouldn't you use Mendeley or another citation manager? Mendeley is free, and you can annotate it as well. See https://www.mendeley.com/reference-management/reference-manager for more information.
At line 197, how will you deconstruct and reconstruct the qualitative data? Will you use a software program, like NVivo, Atlas.ti, or Nudist?
Other than these comments, the methods are well-explained and detailed.
Discussion: well described.
References: not in mdpi style.
Thank you for the opportunity to review your protocol write-up. Best of luck in deriving better PIP definitions.
